# A Machine Learning-Based Applied Prediction Model for Identification of Acute Coronary Syndrome (ACS) Outcomes and Mortality in Patients during the Hospital Stay

**DOI:** 10.3390/s23031351

**Published:** 2023-01-25

**Authors:** Syed Waseem Abbas Sherazi, Huilin Zheng, Jong Yun Lee

**Affiliations:** Department of Computer Science, Chungbuk National University, Cheongju 28644, Chungbuk, Republic of Korea

**Keywords:** machine learning, predictive model, acute coronary syndrome, soft-voting ensemble classifier, imbalanced data, diagnosis and prognosis

## Abstract

Nowadays, machine learning (ML) is a revolutionary and cutting-edge technology widely used in the medical domain and health informatics in the diagnosis and prognosis of cardiovascular diseases especially. Therefore, we propose a ML-based soft-voting ensemble classifier (SVEC) for the predictive modeling of acute coronary syndrome (ACS) outcomes such as STEMI and NSTEMI, discharge reasons for the patients admitted in the hospitals, and death types for the affected patients during the hospital stay. We used the Korea Acute Myocardial Infarction Registry (KAMIR-NIH) dataset, which has 13,104 patients’ data containing 551 features. After data extraction and preprocessing, we used the 125 useful features and applied the SMOTETomek hybrid sampling technique to oversample the data imbalance of minority classes. Our proposed SVEC applied three ML algorithms, such as random forest, extra tree, and the gradient-boosting machine for predictive modeling of our target variables, and compared with the performances of all base classifiers. The experiments showed that the SVEC outperformed other ML-based predictive models in accuracy (99.0733%), precision (99.0742%), recall (99.0734%), F1-score (99.9719%), and the area under the ROC curve (AUC) (99.9702%). Overall, the performance of the SVEC was better than other applied models, but the AUC was slightly lower than the extra tree classifier for the predictive modeling of ACS outcomes. The proposed predictive model outperformed other ML-based models; hence it can be used practically in hospitals for the diagnosis and prediction of heart problems so that timely detection of proper treatments can be chosen, and the occurrence of disease predicted more accurately.

## 1. Introduction

During the last few decades, benefiting from the powerful ability of machine learning and deep learning has achieved great success in health informatics, disease diagnosis and prediction, risk score and healthcare analysis [1,2], specifically in cardiology and cardiovascular disease. With the use of advanced diagnosis and prediction techniques, patients as well as paramedical staff have benefited from the timely detection of proper treatments as well as the severity of the patients [3,4] affected with cardiovascular disease. As these prediction models are dependent on critical risk factors and clinical outcomes of the patients, they can predict the occurrence of diseases more accurately and promptly.

### 1.1. Research Motivation

Scientists and researchers are continuously focusing on the development of diagnosis and prognosis models for their practical use during clinical practice and routine follow-ups [5]. Nevertheless, previous risk detection studies and existing clinical research works mainly focus on either regression-based prediction algorithms, or only deal with a few risk factors while ignoring the tremendous risk factors and correlation between important risk factors [6]. According to the Framingham heart study [7], the prediction models for prediction of heart diseases have mainly been categorized into two groups, such as regression-based risk assessment methods and machine learning-based classification methods. The previous risk assessment methods have been the Framingham Risk Score (FRS) [8,9,10], QRISK [11,12], Thrombolysis in Myocardial Infarction (TIMI) [13,14], Global Registry of Acute Coronary Events (GRACE) [15,16], and History, Electrocardiogram, Age, and Risk factors and Troponin (HEART) [17,18,19] models, whereas the machine learning-based classification methods for heart disease [20] are Random Forest (RF) [21,22,23], Extra Tree (ET) [20,24], Support Vector Machine (SVM) [25], Gradient Boosting Machine (GBM) [26,27], Neural Networks (NN) [28,29,30], and other ensemble models [6,31]. However, there are some drawbacks and limitations of previous models, as follows. Regression-based risk assessment models are outdated and consider only a few risk scores for the early prediction and diagnosis of ACS, whereas already developed machine learning-based risk score models have low accuracy, and those models are specified to only a few risk factors and do not provide accurate predictions for other classes [32]. Furthermore, regression models and decision trees provide limited interpretability, specifically in data nonlinearity [33]. Previous prediction models cannot deal with imbalanced data problems and lead to negative prediction performance, which can lead patients towards mortality [34]. Consequently, most of the previous research studies have focused on the clinical follow-ups of patients while ignoring the hospital stay. The patients are in more medically critical and serious situations during the hospital stay, and discharged from the hospital either after their recovery, or discharged in the case of death or hopeless situations.

### 1.2. Research Objectives

Therefore, this paper proposes a machine learning-based predictive model for the prediction of acute coronary syndrome (ACS) outcomes and mortality of the patients during the hospital stay. The main goal of this research is to develop a machine learning-based predictive model for the prediction of ACS outcomes such as ST-elevation myocardial infarction (STEMI) and non-ST-elevation myocardial infarction (NSTEMI), as well as mortality prediction of in-hospital patients due to heart problems. This research also focuses on the prediction of discharge patients whether the patients have been diagnosed with or recovered from ACS, died during the hospital stay, discharged due to hopelessness by the hospitals, or referred to other hospitals due to medical reasons. Our research content can be summarized as follows. First, we conducted experiments on the Korea Acute Myocardial Infarction Registry (KAMIRNIH) dataset for the prediction of our target outcome, which was highly imbalanced, redundant, and contained missing values as well as irrelevant data regarding the patients’ medical records. We could not directly use this data for the experimental analysis, and thus needed data preprocessing, which deals with data problems such as missing values and data redundancies, drops irrelevant data, and goes through data-balancing techniques so that a proper dataset can be used for predictive modeling. Second, we applied the most-used machine learning models, such as random forest, extra tree, and gradient boosting machine, and combined them to develop an ensemble predictive model for prediction of ACS outcomes and mortality of the patients during the hospital stay. Third, we focused on multiple risk factors and designed the model for the prediction of ACS outcomes (e.g., STEMI, NSTEMI), mortality of patients (e.g., cardiac death, non-cardiac death), and discharge of the patients (e.g., recovered, died, hopeless discharged, referred to the other hospitals). Finally, we compared the performances of machine learning-based classification models with our proposed voting ensemble predictive model in terms of performance measures.

The research contributions and practical aspect of the proposed approach can be summarized as follows:The machine learning-based prediction model is proposed for identification of acute coronary syndrome (ACS) outcomes and mortality in patients during the hospital stay.For the proposed prediction model, the experimental dataset is preprocessed by applying multiple preprocessing, data cleaning, experimental data extraction, and data sampling methods, resulting in 13,104 patients’ records containing 125 important and useful attributes such as basic medical information, past medical history, family medical history, information about medical diagnosis tools, medical findings, PCI information, and initial diagnosis records, and so forth as the final experimental dataset.To deal with the data imbalance issue, we applied the SMOTETomek hybrid sampling technique to overcome the imbalanced class distribution in the experimental dataset.For the experimental analysis, we applied the various machine learning algorithms and finalized the top three algorithms, such as the random forest, extra tree, and gradient boosting machine, and proposed the soft-voting ensemble classifier using these three algorithms as base classifiers for predictive modeling using the training dataset and then evaluated the performance of applied models using 5-fold cross-validation. After that, we applied the test data for the testing of the proposed predictive model and other machine learning models.For analyses of our experimental dataset, we applied various statistical methods such as the Chi-Square test and Analysis of Variance (ANOVA) test.For the evaluation of the proposed soft-voting ensemble classifier and other machine learning algorithms, we applied performance measures such as the AUC, precision, recall, F1-score, accuracy, and confusion matrices.From the experimental results and performance of the proposed predictive model, we conclude that the proposed model outperformed other models and is effective for timely identification of the ACS, as well as helpful for physicians and patients to identify future cardiac events and select the proper treatment for the patients so that the mortality ratio can be reduced in patients with ACS.

## 2. Materials and Methods

### 2.1. Data Source

For the experimental analysis and research studies, we used the Korea Acute Myocardial Infarction Registry (KAMIR) [35] dataset, which is a nationwide registry for Korean patients affected with heart-related diseases. This is registered in 52 hospitals all over Korea and has all patients’ records registered from November 2005 to December 2019. This KAMIR data registry categorized the whole dataset into 4 groups based on the time of registry of the patients. KAMIR-I has all the patients’ data registered between November 2005 to December 2006, KAMIR-II has the patients’ records from January 2007 to January 2008, KAMIR-III (also known as KorMI-I) has all the medical information of the patients between February 2008 to March 2012, KAMIR-IV (KorMI-II) has the data from April 2012 to December 2015, and KAMIR-NIH has all the registered patients’ records from November 2011 to December 2019. The latest data of this registry is KAMIR-NIH, so we used the KAMIR-NIH data for the experimental analysis. KAMIR-NIH consists of 13,104 patients’ records with 2-year follow-ups after hospital discharge. This data contains 551 attributes of the registered patients, such as patients’ past medical history, basic medical information, drugs prescribed and used, rehospitalization history, and cardiac and non-cardiac disease records for the registered patients. This dataset contains the patients’ information during the hospital stay, as well as 6-month, 12-month, and 24-month follow-up records.

### 2.2. Data Extraction and Data Preprocessing

The dataset provided by the KAMIR was in raw form and could not be accessed or used for the experiments, therefore we needed to extract the useful data and preprocess the extracted data according to our experiments and predictive targets. The available dataset was inconsistent, redundant, and contained lots of non-effective information (e.g., follow-ups, date and time, drugs, drugs dosage, treatment methods, and stents information) as well as missing values, so data extraction and preprocessing were the most important steps to use this data for final outcomes of the predictions and diagnosis of ACS for in-hospital patients. The overall data extraction process is briefly elaborated on in Figure 1.

The KAMIR-NIH consists of 13,104 patients’ records having 551 medical attributes. This dataset has basic medical information of the registered patients, patients’ medical history, drugs used for the treatment and drugs dosage, rehospitalization history, cardiac and non-cardiac disease records for the registered patients, in-hospital medical records, as well as 6-month, 12-month, and 24-month follow-up records for the patients. First, we used the KAMIR-NIH raw data and excluded the attributes containing the follow-up information because our research focuses on the patients who were admitted to hospital and stayed for medical examinations regardless of the 6-month, 12-month, and 24-month follow-ups. We excluded 236 attributes containing follow-up information from the total of 551 attributes. In the next step, we omitted the 185 attributes containing unnecessary information such as date and time attributes, drug attributes, drug dosage attributes, treatment methods used during the hospital stays, and attributes containing stents information, such as stent size, diameter, length, and so forth. Furthermore, we also removed all the features containing the information of our final predictions. Our final target variables were ACS outcomes, such as ST-elevation myocardial infarction (STEMI) and non-ST-elevation myocardial infarction (NSTEMI), and mortality prediction of the patients admitted in the hospitals due to heart problems such as CD and NCD, and prediction of discharged patients, such as whether the patients had been diagnosed of or recovered from ACS, died during the hospital stay, discharged due to hopelessness by the hospitals, or referred to other hospitals due to some other medical reason. Therefore, we omitted all those variables which contained this medical information. Finally, we had the 13,104 patients’ records containing 125 important and useful attributes, such as basic medical information (e.g., gender, age, height, weight, heart rate), past medical history (e.g., chest pain, diabetes, hypertension, previous heart failure), family medical history, information about medical diagnosis tools (e.g., electrocardiogram, Image Finding for MI, MRI, CT scan, ECHO), medical findings (e.g., white blood cells, neutrophil, hemoglobin, platelets, glucose, creatinine, cholesterol), PCI information, and initial diagnosis records.

In the KAMIR-NIH dataset, we had different features which could be classified as categorical features, continuous features, and discrete features [36]. For each type of classified feature, we used different preprocessing rules to convert this data so that we could use different machine learning classifiers. First of all, we excluded all the unnecessary attributes from the raw dataset and selected the important features for our implementation. There were lots of missing values in the select attributes, so we implemented multiple missing values’ imputation methods, such as mean value imputation, median, k-nearest neighbours, and also used zero-value imputation to deal with missing values.

This is Example 1 of an equation:(1)$$\mathrm{Mean}\;\mathrm{Value}=\frac{\mathrm{Sum}\;\mathrm{of}\;\mathrm{all}\;\mathrm{values}\;\mathrm{in}\;\mathrm{the}\;\mathrm{colum}}{\mathrm{Total}\;\mathrm{number}\;\mathrm{of}\;\mathrm{values}}$$
(2)$$\mathrm{Median}\;\mathrm{Value}=\left\{\begin{array}{l} \left(\frac{n+1}{2}\right)\mathrm{th}\;\mathrm{value}\;\;\;\;\;\mathrm{if}\;\mathrm{Odd}\;\mathrm{no}.\;\mathrm{of}\;\mathrm{values}\\ \frac{\frac{\mathrm{nth}\;}{2}\;\mathrm{value}+\left(\frac{n}{2}+1\right)\mathrm{th}\;\mathrm{value}}{2}\;\;\;\mathrm{if}\;\mathrm{Even}\;\mathrm{no}.\;\mathrm{of}\;\mathrm{values} \end{array}\right.$$(3)kNN Value = Mean value of k-neighbours of the missing value 

We compared and confirmed that the zero-value imputation methods worked more efficiently than the mean or median value imputation methods, so we finalized the zero-value imputation method to deal with missing values. For categorical features, we used one-hot encoding [37] and label encoding [38] to convert them into numerical form. For continuous variables and discrete variables, we used the actual values of these attributes so that we could minimize data loss. Furthermore, for the variables containing multiple values, we used the one-hot encoding method so that we could use all the possible outcomes of those attributes. For binary-valued attributes, we simply converted them into 1 and 2, so that 1 represented Yes and 2 represented No, whereas if there were missing values, we denoted them with 0.

### 2.3. Data Sampling

Data sampling is an efficient and valuable technique for the transformation of an imbalanced training dataset into balanced class distribution either by increasing the minority class data (oversampling), or decreasing the majority class data (undersampling), or using both strategies at the same time (hybrid sampling). From previous studies of health informatics research and data sampling techniques [39,40,41,42], it has been concluded that hybrid sampling techniques are considered the best data sampling techniques. According to our target variables, our final prediction results include the ACS outcomes along with the discharge reason and death type (final diagnosis + discharge result + death type) for the patients admitted to hospital due to heart problems. In our dataset, our final predictions included 10 target variables, such as “STEMI CD” (N = 290), “STEMI NCD” (N = 38), “STEMI Hopeless Discharge” (N = 24), “STEMI Recovery to Home” (N = 5860), “STEMI Recovery to Other Hospital” (N = 113), “NSTEMI CD” (N = 135), “NSTEMI NCD” (N = 40), “NSTEMI Hopeless Discharge” (N = 25), “NSTEMI Recovery to Home” (N = 6406), and “NSTEMI Recovery to Other Hospital” (N = 173). The data for each target variable in the original dataset are highly imbalanced. We applied the machine learning-based classifiers as well as our proposed predictive model for the final prediction of these target variables, and results were highly biased towards the majority class. We applied the SMOTETomek hybrid sampling technique to balance the class distribution in the experimental dataset. The SMOTETomek hybrid sampling technique uses the SMOTE for oversampling and Tomek Links for data cleaning. This oversampling method does not generate the duplicates of the original dataset. Instead, it creates synthetic data points which are different from the original ones. It moves the data points in the direction of its neighbours so that the synthetic data points are not exactly the same as the original data, but are not completely the same as the original data. Tomek Links in SMOTETomek was used for data cleaning as it removes the majority class data that has less distance from the minority class. This is the reason why we preferred SMOTETomek over the other methods.

### 2.4. Architecture of Proposed Predictive Modeling System

The overall architecture of the proposed predictive modeling system for ACS outcomes and discharge prediction is briefly elaborated in Figure 2. The steps for the predictive modeling system are as follows. In the first step, we used the KAMIR-NIH raw dataset for our experiments and then extracted the useful features from the overall dataset. In the next step, we applied the preprocessing rules on the extracted data to convert the data into a useful form, so that we could apply the different machine learning-based classification models and evaluate the models. We preprocessed the data by applying missing-value imputation techniques, data transformation techniques, and data encoding techniques on our experimental data. After that, we split the data into two main groups, such as training data (70%) and testing data (30%), and then further subdivided the training dataset into a validation dataset using 5-fold cross-validation. In the 5-fold cross-validation, the training dataset was divided into 5 equal sub-groups of datasets and we trained the model using 4 subgroups as training data and the fifth subgroup as test data during the training process. In the next step, the other subgroup was considered as the test dataset and we trained the model on the rest of the 4 subgroups. This process was repeated for 5 iterations until all subgroups were used as test data during the training process of the model. In the next step, we applied the machine learning-based classifiers such as the random forest, extra tree, gradient boosting machine, and a proposed machine learning-based voting ensemble classifier for predictive modeling using the training dataset and then evaluated the performance of the applied models using 5-fold cross-validation. In the next step, we used the test data (30%) for the testing of machine learning-based classification models and the proposed predictive model, and then evaluated the model’s performance using the area under the ROC curve (AUC), accuracy, precision, recall, F-score, and the confusion matrix. Finally, the prediction results were extracted for applied machine learning-based classification models and we classified the extracted results as the final output of our predictive modeling system.

### 2.5. Applied Machine Learning Algorithms

For the experiment, we applied various machine learning algorithms, such as RF, ET, SVM, GBM, GLM, Linear Regression, and Logistic Regression. We compared the results of all applied algorithms and finalized the top 3 algorithms, such as RF, ET, and GBM with outperformed results. The accuracy of these selected ML algorithms was comparatively high, and these were better prediction models for the prescribed outcomes. The other algorithms had lower performance, and therefore we skipped those algorithms and also did not include the other algorithms as base classifiers in the soft-voting ensemble classifier because those algorithms could cause a decrease in the performance of the proposed ensemble classifier. Therefore, we proposed the soft-voting ensemble classifier using RF, ET, and GBM as base classifiers for the identification of acute coronary syndrome (ACS) outcomes and mortality in patients during their hospital stay.

#### 2.5.1. Random Forest

Random forest [21,22,23] is a learning method for classification and regression in which decision trees are constructed such that these trees depend on independent sampled values of a random vector, and the distribution is the same for all trees in random forest. Random forests are constructed randomly, and their results are extracted by combining the decisions of several trees which are trained independently, and final predictions are merged through averaging [28] the prediction trees. When training of the random forest is performed, it can predict the results for new unlabeled data. Prediction is performed identically on each decision tree and random forest finalizes the predictions on the bases of averages of all predictions from each decision tree.

#### 2.5.2. Extra Tree

Extra tree is also known as extremely randomized trees [20,24] in which feature selection and split selection are performed randomly, and it is less computationally expensive as compared with random forest. The key difference between decision trees, random forests, and extra trees is that extra trees show low variance, whereas decision trees and random forest show high variance and medium variance, respectively. It also minimizes over-learning from datasets and controls overfitting, and hence was used in our experiment. The extra tree algorithm yields state-of-the-art results in complex problems of high dimensionality. The decision stumps for extra tree classifiers are built in such a way that all data used in the training dataset are utilized for building each stump, the maximum depth is one, and the best split to form the root node or any node is determined by inquiring into a subset of randomly selected features.

#### 2.5.3. Gradient Boosting Machine

Gradient boosting [26,27] is a powerful machine learning technique for predictive modeling in the form of an ensemble of weak prediction models, generally decision trees. Gradient boosting is a machine learning technique used to solve classification and regression problems in such a way that it strengthens the model with weak predictions to make it better and minimize the loss function such that test loss becomes a minimum. It causes the loss function to be optimized, weak learners to make predictions, and additive models to add weak learners to minimize the loss function. The reason for using this model is that the user-specified cost function can be optimized instead of having a loss function that provides less control, and does not correspond to real-world problems and real-time applications.

### 2.6. Design of a Machine Learning-Based Soft-Voting Ensemble Classifier for Predictive Modeling

In this section, we describe the framework to propose the machine learning-based soft-voting ensemble classifier for the predictive modeling of our results, such as ACS outcomes and discharge reasons. Basically, the voting ensemble classifier relies on the multiple machine learning-based base classifiers which collectively take part in the final prediction results and finalize the predictive results on the bases of the highest weighted probability result. It combines the multiple machine learning models to increase predictive accuracy and minimize the flaws of individual classifiers. The mathematical equation for the prediction of the soft-voting ensemble classifier is as follows:(4)$$P=\arg\underset{i}{\max}\sum_{j=1}^{m}W_{j}f_{c_{A}}\left(C_{j}\left(x\right)=i\right)$$
where C_j_ = classifier; W_j_ = weights associated with the prediction of classifier; $f_{c_{A}}{=\mathrm{characteristic}\;\mathrm{function}\;[C}_{j}\left(x\right)=i\;\in \;A]$; and A = set of class labels.

The design steps of the proposed predictive model are as follows. In the first step, we applied multiple machine learning-based models on our preprocessed data and extracted the results. After that, we selected three machine learning-based classifiers, such as random forest, extra tree, and gradient boosting machine with the best prediction result and dropped other classifiers as their accuracies and performances were comparatively lower than the others. After that, we trained the base classifiers and tuned their hyperparameters. In the next step, we assigned the weights to these base classifiers to increase the performance of our ensemble model and combined the results of these base classifiers using the soft-voting strategy. In the soft-voting ensemble classifier, the final outcome is the result with the highest sum of weight probabilities of the base classifiers. Using the soft-voting ensemble classifier, we could cover up the drawbacks and low performances of the base classifiers. The architecture of machine learning-based soft-voting ensemble classifier for the predictive modeling of ACS outcomes and discharge classification is shown in Figure 3.

### 2.7. Statistical Analysis

For analyses of our experimental dataset, we applied various statistical methods such as the Chi-Square test and Analysis of Variance (ANOVA) test. In our dataset, we had different types of data, such as categorical features and continuous features. To perform the statistical analysis and check their statistical significance, we applied statistical methods and calculated their significance. A chi-square test was used for categorical variables to show their relationship with target variables and interpret the discrepancies between the actual outcomes and expected outcomes. The ANOVA test was used to analyze the mean value differences and influence of independent variables on target variable. We used the ANOVA test because it compares more than two groups to determine the relationship between them. The mathematical representation for the chi-square and ANOVA tests is mentioned below:(5)$$x^{2}=\sum \frac{{\left(O_{i}{\;-\;E}_{i}\right)}^{2}}{E_{i}}$$
(6)$$F=\frac{\frac{\left(\sum_{i=1}^{k}\left(\frac{T_{i}^{2}}{n_{i}}\right)-\left(\frac{G^{2}}{n}\right)\right)}{k\;-\;1}}{\frac{\sum_{i=1}^{k}\sum_{j=1}^{n_{i}}Y_{\mathrm{ij}}^{2}\;-\;\sum_{i=1}^{k}\left(\frac{T_{i}^{2}}{n_{i}}\right)}{n\;-\;k}}$$

In Equation (5), $x^{2}$ represents chi-square, $O_{i}$ observed value, and $E_{i}$ expected value. In Equation (6), $\frac{\left(\sum_{i=1}^{k}\left(\frac{T_{i}^{2}}{n_{i}}\right)-\left(\frac{G^{2}}{n}\right)\right)}{k-1}$ is the mean square due to treatment (MST), $\frac{\sum_{i=1}^{k}\sum_{j=1}^{n_{i}}Y_{\mathrm{ij}}^{2}-\sum_{i=1}^{k}\left(\frac{T_{i}^{2}}{n_{i}}\right)}{n-k}$ is the mean square due to error (MSE), $T_{i}$ is the group total, $n_{i}$ is the number of observation in group i, n is the total number of observations, G is the grand total for all observations, and $Y_{\mathrm{ij}}$ is an observation. All statistical analyses were performed using IBM SPSS Statistics 23 and Microsoft Excel 365.

### 2.8. Evaluation Method and Performance Measures

We categorized our dataset into two main groups, such as training data (70%) and testing data (30%), and then applied 5-fold cross-validation during the training process. We generated each machine learning-based prediction model with the best hyper-parameters, evaluated by 5-fold stratified cross-validation, and then verified by test dataset (30%). For the evaluation of our applied classifiers, we used certain performance measures such as AUC, precision, recall, F1-score, accuracy, and confusion matrices.

The formulas for the above-mentioned performance measures are as follows:(7)$$\mathrm{Precision}=\frac{\mathrm{TP}}{\mathrm{TP}+\mathrm{FP}}$$
(8)$$\mathrm{Recall}=\frac{\mathrm{TP}}{\mathrm{TP}+\mathrm{FN}}$$
(9)$$F1-\mathrm{Score}=2\times \frac{\left(\mathrm{Recall}\;*\mathrm{Precision}\right)}{\left(\mathrm{Recall}+\mathrm{Precision}\right)}$$
(10)$$\mathrm{Accuracy}=\frac{\mathrm{TP}+\mathrm{TN}}{\mathrm{TP}+\mathrm{FP}+\mathrm{FN}+\mathrm{TN}}$$
where TP, TN, FP, and FN denote True-Positive, True-Negative, False-Positive, and False-Negative.

The confusion matrix shown in Table 1 describes the performances of the machine learning-based predictive models, where each row and column represent the instances in the actual class and predicted class, respectively. The FP is a Type 1 error and FN is a Type 2 error.

### 2.9. Implementation Environments

For statistical analysis and data preprocessing, we used the SPSS 18 for Windows (SPSS Inc., Chicago, IL, USA) [43] and MS Excel for Windows (Microsoft Office 365 ProPlus) [44]. For the implementation of our experiments and performance analysis, we used the 64-bit Windows-based operating system with Intel(R) Core (TM) i7-4770 CPU @ 3.40 GHz 3.40 GHz processor and 24 GB random access memory (RAM). We also used the Jupyter Notebook, an open-source web application, for the development and implementation of the predictive model using Python language (Version 3.7.3) [45], Tensor flow [46], Keras (Version 2.3.1), Scikit-learn [47], numpy [48], pandas [49], and imbalanced-learn [50] libraries.

## 3. Results

This chapter will briefly overview and explain the results of our experimental analysis for the predictive modeling of the ACS outcomes and prediction of the discharge reasons for the patients admitted in the hospitals, and also classify all outcomes of our experiments with proper results analysis. We applied widely used machine learning algorithms such as random forest, extra tree, and the gradient boosting machine, and then proposed the machine learning-based soft-voting ensemble classifier using these machine learning classifiers for predictive modeling of the patients during their hospital stay.

### 3.1. Baseline Characteristics

The KAMIR-NIH dataset was preprocessed and subdivided into 2 groups such as ST-elevation myocardial infarction (STEMI) and non-ST-elevation myocardial infarction (NSTEMI). We applied the Chi-Square test for categorical variables to calculate the percentages and frequencies in dataset, and ANOVA test for continuous variables to compute the means and standard deviations of the selected features. The baseline characteristics of all categorical and continuous features are mentioned in Table 2 and Table A1. Most of the features have a *p*-value < 0.001 and <0.05 which indicates that the attributes are statistically significant and have a very low chance of being wrong, whereas some features have a higher p-value indicating that there is a higher chance of being wrong and of it being statistically not significant. In Table 2, features such as Sex, Age, Chest Pain, and so forth have a *p*-value < 0.001, whereas some features such as hsCRP, ST Type, and so forth have a *p*-value < 0.05, which shows that these attributes are statistically significant and have very low chances of a wrong result. In contrast, features such as Neutrophil, Troponin T, High Density Lipoprotein, and so forth have a higher *p*-value, indicating that these attributes are not statistically significant and have a higher chance of being wrong.

### 3.2. Results of Performance Measures for Applied Machine Learning-Based Predictive Models

For the performance evaluation of applied machine learning-based predictive models for ACS occurrences and discharge results, we compared all applied models using accuracy, precision, recall, F1-score, and AUC as shown in Table 3, Table 4, Table 5 and Table 6. Table 3 shows the detailed results of performance measures for the random forest classifier for all required outcomes of our class labels such as “STEMI CD”, “STEMI NCD”, “STEMI Hopeless Discharge”, “STEMI Recovery to Home”, “STEMI Recovery to Other Hospital”, “NSTEMI CD”, “NSTEMI NCD”, “NSTEMI Hopeless Discharge”, “NSTEMI Recovery to Home”, and “NSTEMI Recovery to Other Hospital”, Table 4 for an extra tree classifier, Table 5 for a gradient boosting machine classifier, and Table 6 for our proposed soft-voting ensemble classifier for predictive modeling of the above-mentioned class labels.

Table 7 shows the overall performance evaluation and comparison of all applied machine learning-based models, such as random forest, extra tree, and gradient boosting machine with our proposed soft-voting ensemble classifier for predictive modeling of ACS outcomes and discharge results for the patients admitted to the hospital. Table 7 presents the overall results of all applied machine learning-based predictive models as the average of all 10 class labels mentioned above. The average accuracies were 98.9172%, 98.9796%, 78.6517%, and 99.0733% for random forest, extra tree, the gradient boosting machine, and the soft-voting ensemble model, respectively. Consequently, the AUC, precision, recall, and f1-score values for random forest were (99.9650%, 98.9182%, 98.9172%, 98.9159%), extra tree (99.9828%, 98.9828%, 98.9797%, 98.9791%), gradient boosting machine (97.7566%, 78.7202%, 78.6517%, 78.6302%), and the soft-voting ensemble classifier (99.9702%, 99.0742%, 99.0734%, 99.9719%), respectively.

From the results of overall performance evaluations for applied machine learning-based predictive models mentioned in Table 7, we could find that the proposed soft-voting ensemble classifier outperformed other predictive models in accuracy, precision, recall, and f1-score, but the extra tree outperformed others in AUC. Overall, the proposed machine learning-based soft-voting ensemble classifier improved the predictive results for our target variables using KAMIR-NIH data for the patients admitted in hospitals due to heart problems.

Figure 4 shows the confusion matrix for the performance evaluation of the proposed soft-voting ensemble classifier using in-hospital data. The confusion matrix was used for performance evaluation in tabular form in which the x-axis and y-axis represent class labels such as “STEMI CD”, “STEMI NCD”, “STEMI Hopeless Discharge”, “STEMI Recovery to Home”, “STEMI Recovery to Other Hospital”, “NSTEMI CD”, “NSTEMI NCD”, “NSTEMI Hopeless Discharge”, “NSTEMI Recovery to Home”, and “NSTEMI Recovery to Other Hospital”. These class labels are denoted as numbers from 0–9 for these 10 class labels, respectively.

Figure 4a represents the actual confusion matrix with classified class labels and misclassified class labels, whereas Figure 4b represents the normalized confusion matrix. The diagonal values in Figure 4a,b represent the accurately classified results and other values misclassified using the machine learning-based soft-voting ensemble classifier.

## 4. Discussion

After the fast revolution of technology in the fields of computer science and health informatics, the practical use of artificial intelligence technologies in cardiology has also been increasing, specifically for the diagnosis and prognosis of ACS occurrences in patients with cardiovascular disease. Therefore, our research focused on the development of the machine learning-based soft-voting ensemble model for the predictive modeling of occurrences of ACS outcomes, hospital discharge for the patients with ACS, and type of death in cases where the patient died during their hospital stay. From the experimental results of machine learning classifiers, we realized that the applied algorithms were accurately predicting a few class labels, but the accuracy was low for other class labels. Therefore, we proposed the machine learning-based soft-voting ensemble classifier for predictive modeling of our target variables. Consequently, we compared the results of applied machine-learning classifiers with the proposed soft-voting ensemble predictive model on the basis of performance measures such as accuracy, precision, recall, F1-score, and AUC. The results showed that the proposed predictive model outperformed the other machine learning classifiers. The performance of the proposed soft-voting ensemble model in accuracy, precision, recall, F1-score, and AUC was 99.0733%, 99.0742%, 99.0734%, 99.9719%, and 99.9702%, as well as random forest (98.9172, 98.9182, 98.9172, 98.9159, 99.9650), extra tree (98.9796, 98.9828, 98.9797, 98.9791, 99.9828), and gradient boosting machine (78.6517, 78.7202, 78.6517, 78.6302, 97.7566), respectively. The overall performance of our predictive model was comparatively higher than other applied machine learning models in accuracy, precision, recall, and F1-score, but the AUC of the extra tree classifier was slightly higher than the proposed predictive model.

## 5. Conclusions

This paper proposed a machine learning-based soft-voting ensemble classifier for the predictive modeling of ACS occurrences (e.g., STEMI, NSTEMI), hospital discharge reports (e.g., Death, Hopeless Discharge, Recovery to Home, Recovery to Other Hospital), and the death type (e.g., Cardiac Death, Non-Cardiac Death) of patients admitted to hospitals using three widely used machine-learning classifiers named as random forest, extra tree, and the gradient boosting machine. Consequently, we can summarize our main contributions as follows. First, this paper led to the development of machine learning-based predictive modeling for the identification of acute coronary syndrome (ACS) outcomes and mortality in patients during their hospital stay. Second, this model can forecast the occurrences of mortality during hospital stays of Korean patients with acute coronary syndrome, because they well-reflect the demographic characteristics of Koreans. Third, it was shown that the performance of the machine learning-based mortality prediction model was superior to previous risk assessment approaches. Lastly, it is expected that these results can contribute to the development of a future diagnosis and forecast tool of the occurrences of major adverse cardiovascular events (MACE) in clinical patients with acute coronary syndrome. The proposed machine learning-based soft-voting ensemble classifier is effective for timely identification of ACS, and is also helpful for the physicians and patients to identify future cardiac events and select the proper treatment for the patients so that the mortality ratio can be reduced in patients with ACS. This research can be used as a satisfactory privilege for the development of a machine learning-based risk score system for patients admitted to hospitals, as well as discharged patients in the near future.

There were some potential limitations to our research. First, we used only 8227 experimental subjects, and thus our dataset is insufficient, because machine learning algorithms need to employ a large-scale dataset in the experiment. Second, our proposed model is also limited in diagnosing and forecasting mortality in Korean patients with acute coronary syndrome. Third, it is difficult to explain the prediction result in machine learning-based approaches, because GBM, RF, and ET were non-linear models, whereas in the regression-based prediction models, it was easy to explain how major prognostic factors were associated with mortality in patients with ACS, because they were based on statistical analysis. Lastly, there was a limit on checking the mortality predictions during hospital stays in our experimental dataset.

## Figures and Tables

**Figure 1 sensors-23-01351-f001:**
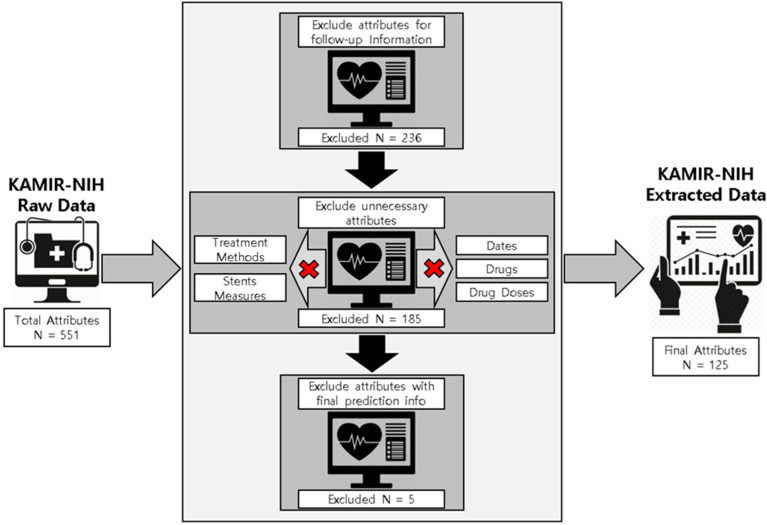
Experimental data extraction from the original KAMIR-NIH dataset.

**Figure 2 sensors-23-01351-f002:**
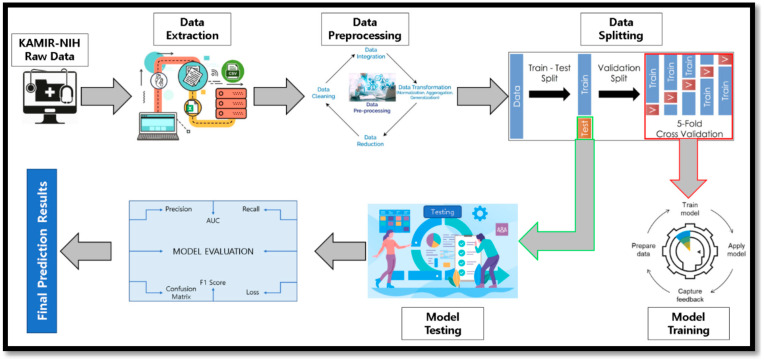
Overall workflow of proposed predictive modeling system.

**Figure 3 sensors-23-01351-f003:**
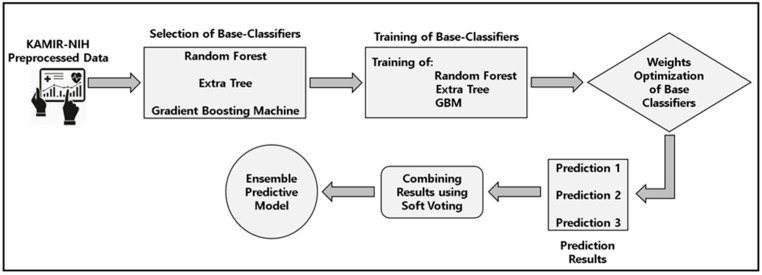
The soft-voting ensemble classifier for predictive modeling of ACS outcomes and mortality of patients.

**Figure 4 sensors-23-01351-f004:**
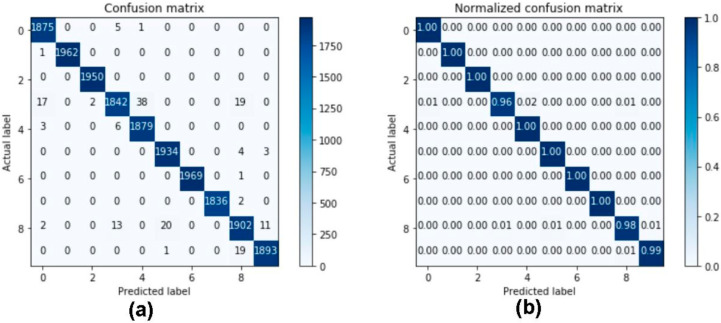
Confusion matrix for all class labels using proposed soft-voting ensemble classifier on KAMIR-NIH in-hospital data (**a**) actual confusion matrix; (**b**) normalized confusion matrix.

**Table 1 sensors-23-01351-t001:** Confusion matrix for predictive modeling of target variables.

	**Predicted Value** (Predicted by the test)		
Actual Value (Confirmed by Experience)		**Positives**	**Negatives**
	**Positives**	**TP** (True Positive)	**FN** (False Negative)
	**Negatives**	**FP** (False Positive)	**TN** (True Negative)

**Table 2 sensors-23-01351-t002:** Baseline characteristics of important features for all subjects (N = 13,104).

Variable	Descriptive Statistics				
	Value	All (N = 13,104)	STEMI (N = 6325)	NSTEMI (N = 6779)	*p* Value
Sex	Male	73.92 (9686)	77.71 (4915)	70.38 (4771)	**<0.001 ****
	Female	26.08 (3418)	22.29 (1410)	29.62 (2008)	**<0.001 ****
Age	Number	63.96925 ± 12.64313	62.745 ± 12.8054	65.112 ± 12.3820	**<0.001 ****
Chest Pain	Typical	86.19 (11,294)	90.85 (5746)	81.84 (5548)	**<0.001 ****
Systolic Blood Pressure	Number	130.1542 ± 30.0512	125.45 ± 31.529	134.54 ± 27.903	**<0.001 ****
Diastolic Blood Pressure	Number	78.64339 ± 18.34411	76.62 ± 19.870	80.53 ± 16.580	**<0.001 ****
Heart Rate	Number	78.6606 ± 19.59055	77.11 ± 20.545	80.11 ± 18.542	**<0.001 ****
Killip Class	III–IV	13.35 (1750)	15.68 (992)	11.18 (758)	**<0.001 ****
Height	Number	164.2221 ± 11.22179	164.950 ± 12.1186	163.548 ± 10.2780	**<0.001 ****
Weight	Number	65.28827 ± 12.1705	66.161 ± 12.0610	64.481 ± 12.2165	**<0.001 ****
Prev. MI	Yes	7.85 (1029)	5.98 (378)	9.60 (651)	**<0.001 ****
Prev. Angina	Yes	9.76 (1279)	6.48 (410)	12.82 (869)	**<0.001 ****
Prev. Heart Failure	Yes	1.63 (213)	0.89 (56)	2.32 (157)	**<0.001 ****
Prev. CVA	Infarction	5.10 (668)	4.05 (256)	6.08 (412)	**<0.001 ****
Smoking	Current	39.02 (5113)	44.08 (2788)	34.30 (2325)	**<0.001 ****
White Blood Cells	Number	10.50721 ± 4.552223	11.529 ± 4.1662	9.553 ± 4.6885	**<0.001 ****
Neutrophil	Number	66.48509 ± 15.18227	66.324 ± 16.2682	66.635 ± 14.0928	0.242
Lymphocyte	Number	24.63949 ± 13.05661	25.337 ± 14.3136	23.989 ± 11.7264	**<0.001 ****
Hemoglobin	Number	13.76865 ± 2.142532	14.127 ± 2.0190	13.435 ± 2.1999	**<0.001 ****
Platelets	Number	232.3399 ± 67.78979	235.89 ± 67.208	229.02 ± 68.166	**<0.001 ****
Glucose	Number	169.9549 ± 82.53465	179.21 ± 82.991	161.40 ± 81.184	**<0.001 ****
creatinine	Number	1.134202 ± 1.2009	1.047 ± 0.8031	1.215 ± 1.4741	**<0.001 ****
Max. Creatine Kinase	Number	1019.191 ± 1955.354	1429.309 ± 2370.662	625.581 ± 1335.430	**<0.001 ****
Creatine Kinase MB	Number	110.5776 ± 164.4305	166.2745 ± 189.54721	58.4153 ± 114.40263	**<0.001 ****
Troponin I	Number	46.83087 ± 105.5838	75.4137 ± 138.28379	21.8091 ± 53.17410	**<0.001 ****
Troponin T	Number	14.21992 ± 459.5798	4.8386 ± 12.85030	26.5103 ± 698.40787	0.306
Total Cholesterol	Number	177.8673 ± 46.35988	180.84 ± 46.078	175.08 ± 46.454	**<0.001 ****
High Density Lipoprotein	Number	42.83839 ± 12.51687	42.61 ± 12.289	43.05 ± 12.727	0.053
Low Density Lipoprotein	Number	111.849 ± 40.58414	114.11 ± 40.616	109.72 ± 40.442	**<0.001 ****
Triglyceride	Number	134.5005 ± 120.0775	140.29 ± 125.358	128.98 ± 114.547	**<0.001 ****
hsCRP	Number	1.554095 ± 6.201701	1.3933 ± 4.38070	1.7238 ± 7.66500	**<0.05 ***
NTproBNP	Number	2795.943 ± 9980.085	1872.666 ± 9574.2647	3695.017 ± 10,281.7662	**<0.001 ****
BNP	Number	319.4587 ± 741.2533	222.417 ± 533.8414	431.961 ± 912.7885	**<0.001 ****
HbA1c	Number	6.486941 ± 1.482016	6.469 ± 1.5192	6.505 ± 1.4450	0.262
ARU	Number	460.0607 ± 73.9739	457.53 ± 75.363	462.30 ± 72.673	0.090
PRU	Number	199.4301 ± 109.9491	181.66 ± 108.618	217.63 ± 108.344	**<0.001 ****
PreTIMI	TIMI 0	42.11 (5518)	61.75 (3906)	23.78 (1612)	**<0.001 ****
	TIMI I	9.81 (1286)	10.29 (651)	9.37 (635)	**<0.001 ****
	TIMI II	13.84 (1813)	10.34 (654)	17.10 (1159)	**<0.001 ****
	TIMI III	23.94 (3137)	14.37 (909)	13.41 (909)	**<0.001 ****
PostTIMI	TIMI 0	0.37 (49)	0.43 (27)	0.32 (22)	**<0.001 ****
	TIMI I	0.43 (57)	0.62 (39)	0.27 (18)	**<0.001 ****
	TIMI II	2.43 (319)	3.49 (221)	1.45 (98)	**<0.001 ****
	TIMI III	86.45 (11,329)	92.22 (5833)	81.07 (5496)	**<0.001 ****
Initial Diagnosis	STEMI	48.08 (6300)	98.78 (6248)	0.77 (52)	**<0.001 ****
	NSTEMI	51.91 (6802)	1.20 (76)	99.22 (6726)	**<0.001 ****
ST Type	Acute ST	0.18 (23)	0.27 (17)	0.09 (6)	**<0.05 ***

Note: hsCRP denotes high-sensitivity C-reactive protein; NTproBNP N-terminal prohormone of brain natriuretic peptide; BNP B-type natriuretic peptide; HbA1c Hemoglon A1c; ARU Aspirin Reaction Unit; PRU P2Y12 Reaction Unit. Note: The single asterisk (*) with p-value denotes that variables are statistically significant as a *p*-value < 0.05 means that there is a less than 5% chance of being wrong and a double asterisk (**) with a p-value denotes that variables are statistically highly significant as a *p*-value < 0.001 means that there is less than one in a thousand chance of being wrong in STEMI and NSTEMI groups.

**Table 3 sensors-23-01351-t003:** Performance measures of all class labels for the Random Forest Classifier using in-hospital data.

Class Labels for Prediction	Precision	Recall	F1-Score	AUC
STEMI CD	0.9862	0.9928	0.9895	0.9999
STEMI NCD	1.0000	0.9994	0.9997	1.0000
STEMI Hopeless Discharge	1.0000	0.9994	0.9997	1.0000
STEMI Recovery to Home	0.9778	0.9576	0.9676	0.9974
STEMI Recovery to other Hospital	0.9818	0.9890	0.9853	0.9999
NSTEMI CD	0.9852	0.9948	0.9900	0.9999
NSTEMI NCD	1.0000	0.9989	0.9994	1.0000
NSTEMI Hopeless Discharge	1.0000	0.9994	0.9997	1.0000
NSTEMI Recovery to Home	0.9670	0.9743	0.9706	0.9981
NSTEMI Recovery to other Hospital	0.9930	0.9851	0.9890	0.9999

**Table 4 sensors-23-01351-t004:** Performance measures of all class labels for Extra Tree Classifier using in-hospital data.

Class Labels for Prediction	Precision	Recall	F1-Score	AUC
STEMI CD	0.9860	0.9956	0.9908	0.9999
STEMI NCD	1.0000	0.9963	0.9981	1.0000
STEMI Hopeless Discharge	0.9994	0.9979	0.9986	1.0000
STEMI Recovery to Home	0.9724	0.9567	0.9645	0.9983
STEMI Recovery to other Hospital	0.9789	0.9839	0.9814	0.9998
NSTEMI CD	0.9930	0.9962	0.9946	0.9999
NSTEMI NCD	0.9994	0.9994	0.9994	1.0000
NSTEMI Hopeless Discharge	1.0000	0.9989	0.9994	0.9999
NSTEMI Recovery to Home	0.9709	0.9874	0.9791	0.9992
NSTEMI Recovery to other Hospital	0.9994	0.9867	0.9930	0.9998

**Table 5 sensors-23-01351-t005:** Performance measures of all class labels for the Gradient Boosting Machine Classifier using in-hospital data.

Class Labels for Prediction	Precision	Recall	F1-Score	AUC
STEMI CD	0.7166	0.7166	0.7166	0.9645
STEMI NCD	0.8263	0.8054	0.8157	0.9881
STEMI Hopeless Discharge	0.7815	0.7150	0.7467	0.9790
STEMI Recovery to Home	0.8504	0.8649	0.8576	0.9853
STEMI Recovery to other Hospital	0.7039	0.7544	0.7283	0.9645
NSTEMI CD	0.7709	0.7442	0.7573	0.9734
NSTEMI NCD	0.8033	0.8922	0.8454	0.9886
NSTEMI Hopeless Discharge	0.8394	0.8045	0.8215	0.9882
NSTEMI Recovery to Home	0.8731	0.8575	0.8653	0.9869
NSTEMI Recovery to other Hospital	0.6996	0.7025	0.7010	0.9525

**Table 6 sensors-23-01351-t006:** Performance measures of all class labels for the proposed Soft Voting Ensemble Classifier using in-hospital data.

Class Labels for Prediction	Precision	Recall	F1-Score	AUC
STEMI CD	0.9882	0.9948	0.9915	0.9999
STEMI NCD	1.0000	0.9984	0.9992	1.0000
STEMI Hopeless Discharge	1.0000	1.0000	1.0000	1.0000
STEMI Recovery to Home	0.9799	0.9596	0.9696	0.9977
STEMI Recovery to other Hospital	0.9792	0.9860	0.9826	0.9998
NSTEMI CD	0.9879	0.9989	0.9933	0.9999
NSTEMI NCD	0.9989	0.9994	0.9992	1.0000
NSTEMI Hopeless Discharge	1.0000	0.9979	0.9989	1.0000
NSTEMI Recovery to Home	0.9754	0.9826	0.9790	0.9989
NSTEMI Recovery to other Hospital	0.9974	0.9904	0.9939	0.9998

**Table 7 sensors-23-01351-t007:** Overall performance evaluation for all applied predictive models during in-hospital stays using KAMIR-NIH in-hospital data.

Algorithms	Accuracy	AUC	Precision	Recall	F1-Score
Random Forest	98.9172	99.9650	98.9182	98.9172	98.9159
Extra Tree	98.9796	**99.9828**	98.9828	98.9797	98.9791
Gradient Boosting Machine	78.6517	97.7566	78.7202	78.6517	78.6302
Proposed SVE Classifier	**99.0733**	99.9702	**99.0742**	**99.0734**	**99.9719**

## Data Availability

The experimental data can be accessed and available on special request to the Korea Acute Myocardial Infarction Registry (KAMIR) (http://kamir5.kamir.or.kr/).

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
