# Peer review of "A Machine Learning-Based Applied Prediction Model for Identification of Acute Coronary Syndrome (ACS) Outcomes and Mortality in Patients during the Hospital Stay"

_sensors, 2023, doi:10.3390/s23031351_

Round 1

Reviewer 1 Report

This paper presents an ensemble ML classifier on acute coronary syndrome outcomes, which uses the soft-voting technique on multiple ML algorithms such as random forest, extra tree, GBM. The authors have provided details on the dataset, the preprocessing techniques, the results of the proposed SVEC system and a complete statistical analysis of the features.

While I believe that this paper has a quite positive impact to the research community, it also comes with several drawbacks that will require a major revision to deal with.

Here are the detailed comments:
- The motivation of this research is quite clear.
- The paper is well-organized. I can clearly understand the idea behind each section.
- The experiments are correctly conducted. The authors have explained clearly their data preprocessing method and the reason behind the choices  (e.g. dealing with missing values, unbalanced dataset,...)
- Multiple experiments are presented to highlight the effectiveness of the proposed approach on a binary classification task (STEMI and NSTEMI), and a multiclass task.

Major drawback:
- While this paper highlights the accuracy of the proposed SVEC method and has a comparison between the SVEC approach and each ML technique that contributed to the voting process, I believe it is crucial for this kind of paper to compare the SVEC approach with other ML/DL/Statistical approaches. In this paper, the authors only compared the SVEC approach with RF, Extra tree and GBM (which are the 3 algorithms in SVEC), thus make the comparison quite insignificant to readers.

Minor issues:
- The paper has to be proofread because some sentences are hard to read (mostly due to incorrect grammar)
For example:
The dataset provided by the KAMIR is in raw form and cannot be accessed or used for the experiments, therefore it needs to extract the useful data and preprocess the extracted data according to our experiments and predictive targets.
I think "it needs to extract" has to be "we need to extract"
Same for multiple other sentences.
- In section 2.2 the authors have listed several "useless" information that are removed from the dataset such as drugs, drugs dosage, treatment used, stent information.... I would like to have a clearer explanation to that removal.

Author Response

Manuscript ID: sensors-2171136

Manuscript Title: “A machine learning applied prediction model for identification of acute coronary syndrome (ACS) outcomes and mortality in patients during the hospital stay”

To: Sensors Editor

Re: Response to reviewers

Dear Ms. Charlotte Li and Respected Reviewers,

Thank you and your team for considering our manuscript, entitled: “A machine learning applied prediction model for identification of acute coronary syndrome (ACS) outcomes and mortality in patients during the hospital stay” and the tireless efforts spent in reviewing the paper.

We are thankful to you and reviewers for the perceptive comments and valuable suggestions to further improve our manuscript. We have complied with all the requirements in detail. We hope that, with these modifications and improvements based on the comments, the quality of our manuscript would meet the publication standard of Sensors.

We are uploading our (a) point-by-point response to the comments (below) (response to reviewers), (b) an updated manuscript with yellow highlighting indicating changes, and (c) a clean updated manuscript without highlights.

Thank you very much for your kind consideration.

Best regards,

Prof. Jong Yun LEE

Department of Computer Science,

Chungbuk National University (CBNU),

Cheongju Chungbuk, 28644 Korea.

Response to Reviewers’ Comments

Reviewer 1

This paper presents an ensemble ML classifier on acute coronary syndrome outcomes, which uses the soft-voting technique on multiple ML algorithms such as random forest, extra tree, GBM. The authors have provided details on the dataset, the preprocessing techniques, the results of the proposed SVEC system and a complete statistical analysis of the features.

While I believe that this paper has a quite positive impact to the research community, it also comes with several drawbacks that will require a major revision to deal with.

Here are the detailed comments:

- The motivation of this research is quite clear.

- The paper is well-organized. I can clearly understand the idea behind each section.

- The experiments are correctly conducted. The authors have explained clearly their data preprocessing method and the reason behind the choices (e.g. dealing with missing values, unbalanced dataset,...)

- Multiple experiments are presented to highlight the effectiveness of the proposed approach on a binary classification task (STEMI and NSTEMI), and a multiclass task.

Major drawback:

- While this paper highlights the accuracy of the proposed SVEC method and has a comparison between the SVEC approach and each ML technique that contributed to the voting process, I believe it is crucial for this kind of paper to compare the SVEC approach with other ML/DL/Statistical approaches. In this paper, the authors only compared the SVEC approach with RF, Extra tree and GBM (which are the 3 algorithms in SVEC), thus make the comparison quite insignificant to readers.

Response: Thank you so much for your valuable comment and suggestion to compare the experimental results with other algorithms as well. We really appreciate your efforts to read our manuscript carefully and give us reviews to improve the quality of the manuscript. For the experiment, we applied multiple machine learning algorithms such as Random Forest, Extra Tree, Support Vector Machine (SVM), Gradient Boosting Machine (GBM), Generalized Linear Model (GLM), Linear Regression, and Logistic Regression. We compared the results of all applied algorithms and choose the top 3 algorithms such as Random Forest, Extra Tree, and Gradient Boosting Machine (GBM) with outperformed results. The other algorithms had very low experimental results (less than 70%) therefore we skipped those algorithms from this manuscript as well as we did not include the other algorithms in soft voting ensemble classifiers as those algorithms could cause the decreasing of the performance of the proposed ensemble classifier. Therefore, we proposed the soft voting ensemble classifier as predictive model for the prescribed outcomes using Random Forest, Extra Tree, and Gradient Boosting Machine as base classifiers and compared the results of the base classifiers with the proposed SVEC using performance measures such as precision, recall, F1-score, AUC, and accuracy. The experimental results have verified the performance and authenticity of the proposed soft voting ensemble classifier for the predictive modelling of the target variables.

Minor issues:

- The paper has to be proofread because some sentences are hard to read (mostly due to incorrect grammar)

For example:

The dataset provided by the KAMIR is in raw form and cannot be accessed or used for the experiments, therefore it needs to extract the useful data and preprocess the extracted data according to our experiments and predictive targets.

I think "it needs to extract" has to be "we need to extract"

Same for multiple other sentences.

Response: Thank you for your comments and suggestions to improve the quality of the manuscript. In the revised version of the manuscript, we have updated by proofreading all chapters of the manuscript and adding the contents as well as revising the contents as per your suggestions. The modifications are highlighted in the revised version of the manuscript. Please have a look. The updates are highlighted in yellow colour.

- In section 2.2 the authors have listed several "useless" information that are removed from the dataset such as drugs, drugs dosage, treatment used, stent information.... I would like to have a clearer explanation to that removal.

Despite the satisfactory quality of the article, some shortcomings need to be corrected.

Response: We are thankful for your efforts to review the manuscript and highlight the important issues in the manuscript. The KAMIR-NIH consists of 13,104 patients record having 551 medical attributes. This dataset has basic medical information of the registered patients, patients’ medical history, drugs used for the treatment and drugs dosage, rehospitalization history, cardiac and non-cardiac disease records for the registered patients, in-hospital medical records as well as 6-month, 12-month, and 24-month follow-up records for the patients. First, we excluded the attributes containing the follow-ups information because our research focuses on the patients admitted in the hospital and stayed for medical examination regardless of the 6-month, 12-month, and 24-month follow-ups. We excluded 236 attributes containing follow-ups information from the total of 551 attributes. In the next step, we omitted the 185 attributes containing the unnecessary information such as date and time attributes, drugs attributes, drugs dosage attributes, treatment methods used during the hospital stays, and attributes containing the stents information such as stents size, diameter, length etc. Furthermore, we also removed all the features contains the information of our final predictions. As our final target variables are ACS outcomes such as ST-elevation myocardial infarction (STEMI) and non-ST-elevation myocardial infarction (NSTEMI), mortality prediction of the patients admitted in the hospitals due to heart problem such as CD and NCD, and prediction of discharge patients whether the patients are diagnosed and recovered from the ACS, died during the hospital stay, discharged hopelessly from the hospitals, or referred to the other hospitals due to some medical reasons. Therefore, we omitted all those variables which contains this medical information. Finally, we have the 13,104 patients’ record containing 125 important and useful attributes such as basic medical information (e.g., gender, age, height, weight, heart rate etc.), past medical history (e.g., chest pain, diabetes, hypertension, previous heart failure etc.), family medical history, information about medical diagnosis tools (e.g., electrocardiogram, Image Finding for MI, MRI, CT scan, ECHO etc.), medical findings (e.g., white blood cells, neutrophil, haemoglobin, platelets, glucose, creatinine, cholesterol etc.), PCI information, and initial diagnosis records etc.

We have mentioned the detailed explanation of the Data Extraction and Data Pre-processing in the updated version of the manuscript.

Reviewer 2 Report

Comment1 1: The author should revise and improve the English quality of the paper.

Comment 2: The author used the synthetic minority oversampling technique (SMOTE) to overcome the data imbalance problem. Why didn’t he use the other oversampling and undersampling methods? Please justify.

Comment 3: Did the authors apply the other machine learning algorithms to the experimental dataset? If so, why didn’t you include their results in the manuscript?

Comment 4: The authors used the KAMIR-NIH dataset for the experiment. Is this publicly available dataset? If it is not available, give the source of the dataset and official website for the request of data. You have to provide the dataset and the manuscript as supplementary material.

Comment 5: In Figure 1, why did you remove the attributes such as treatment methods, stents, drugs, etc.? Please justify.

Comment 6: Why didn’t you use the t-test to verify the significance and statistical analysis of the experimental dataset?

Comment 7: As mentioned in Table 2, Baseline Characteristics, some attributes have a high p-value and are statistically insignificant. Why did you include these attributes in your experiment?

Comment 8: The research limitations should be merged with the conclusion and removed the section 5.1.

Author Response

Manuscript ID: sensors-2171136

Manuscript Title: “A machine learning applied prediction model for identification of acute coronary syndrome (ACS) outcomes and mortality in patients during the hospital stay”

To: Sensors Editor

Re: Response to reviewers

Dear Ms. Charlotte Li and Respected Reviewers,

Thank you and your team for considering our manuscript, entitled: “A machine learning applied prediction model for identification of acute coronary syndrome (ACS) outcomes and mortality in patients during the hospital stay” and the tireless efforts spent in reviewing the paper.

We are thankful to you and reviewers for the perceptive comments and valuable suggestions to further improve our manuscript. We have complied with all the requirements in detail. We hope that, with these modifications and improvements based on the comments, the quality of our manuscript would meet the publication standard of Sensors.

We are uploading our (a) point-by-point response to the comments (below) (response to reviewers), (b) an updated manuscript with yellow highlighting indicating changes, and (c) a clean updated manuscript without highlights.

Thank you very much for your kind consideration.

Best regards,

Prof. Jong Yun LEE

Department of Computer Science,

Chungbuk National University (CBNU),

Cheongju Chungbuk, 28644 Korea.

Response to Reviewers’ Comments

Reviewer 2

Comment 1: The author should revise and improve the English quality of the paper.

Response 1: We are thankful for your valuable comments. We really appreciate your efforts to read our manuscript carefully and give us reviews to improve the quality of the manuscript. We have updated the manuscript by proofreading all chapters of the manuscript and adding the contents as well as revising the contents in the updated version of our paper. The modifications are highlighted in the revised version of the manuscript.

Comment 2: The author used the synthetic minority oversampling technique (SMOTE) to overcome the data imbalance problem. Why didn’t he use the other oversampling and undersampling methods? Please justify.

Response 2: Thank you so much for highlighting the important point to update the manuscript. In the experiment, the experimental dataset was highly imbalanced. Therefore, we applied the synthetic minority oversampling technique (SMOTE) to overcome the data imbalance problem. There are multiple variants of SMOTE, one of the best method is SMOTETomek which uses the SMOTE for oversampling and Tomek Links for data cleaning. This oversampling method does not generate the duplicates of the original dataset. Instead, it creates the synthetic data points which are different from the original ones. It moves the data points in the direction of its neighbours so that the synthetic data points are not exactly same as original data, as well as not completely from the original data. Tomek Links in SMOTETomek is used for data cleaning in a way that it removes the majority class data that has less distance from the minority class.

This is the reason to prefer SMOTETomek over the other methods.

The process of SMOTE-Tomek Links is as follows.

  1. (Start of SMOTE) Choose random data from the minority class.
  2. Calculate the distance between the random data and its k nearest neighbors.
  3. Multiply the difference with a random number between 0 and 1, then add the result to the minority class as a synthetic sample.
  4. Repeat step number 2–3 until the desired proportion of minority class is met. (End of SMOTE)
  5. (Start of Tomek Links) Choose random data from the majority class.
  6. If the random data’s nearest neighbor is the data from the minority class (i.e. create the Tomek Link), then remove the Tomek Link.

We have also added the detailed explanation of SMOTETomek in the updated manuscript. Please have a look in section “2.3. Data Sampling”. The updates are highlighted in yellow colour.

Comment 3: Did the authors apply the other machine learning algorithms to the experimental dataset? If so, why didn’t you include their results in the manuscript?

Response 3: Thank you so much for your valuable comment and suggestion to compare the experimental results with other algorithms as well. For the experiment, we applied various machine learning algorithms such as Random Forest, Extra Tree, Support Vector Machine (SVM), Gradient Boosting Machine (GBM), Generalized Linear Model (GLM), Linear Regression, and Logistic Regression. We compared the results of all applied algorithms and choose the top 3 algorithms such as Random Forest, Extra Tree, and Gradient Boosting Machine (GBM) with outperformed results. The other algorithms had lower experimental results (less than 70%) therefore we skipped those algorithms from this manuscript as well as we did not include the other algorithms in soft voting ensemble classifiers as those algorithms could cause the decreasing of the performance of the proposed ensemble classifier. Therefore, we proposed the soft voting ensemble classifier as predictive model for the prescribed outcomes using Random Forest, Extra Tree, and Gradient Boosting Machine as base classifiers and compared the results of the base classifiers with the proposed SVEC using performance measures such as precision, recall, F1-score, AUC, and accuracy. The experimental results have verified the performance and authenticity of the proposed soft voting ensemble classifier for the predictive modelling of the target variables.

We have included the detailed explanation of the applied machine learning algorithms in the updated version of the manuscript. Please see the section “2.5. Applied Machine Learning Algorithms” in the updated manuscript. The updates are highlighted in yellow colour.

Comment 4: The authors used the KAMIR-NIH dataset for the experiment. Is this publicly available dataset? If it is not available, give the source of the dataset and official website for the request of data. You have to provide the dataset and the manuscript as supplementary material.

Response 4: Thank you so much for your comment. The dataset is provided by the Korea Acute Myocardial Infarction Registry (KAMIR-NIH) for experimental analysis. We are not allowed to provide the experimental dataset as the supplementary material for the research article as the KAMIR has imposed the ethical restriction to share the dataset with others. The experimental datasets are confidential and available with the permission of Korea Acute Myocardial Infarction Registry KAMIR (http://kamir5.kamir.or.kr/) on reasonable request.

Comment 5: In Figure 1, why did you remove the attributes such as treatment methods, stents, drugs, etc.? Please justify.

Response 5: Thank you so much for highlighting the important point to update the manuscript. The KAMIR-NIH consists of 13,104 patients record having 551 medical attributes. This dataset has basic medical information of the registered patients, patients’ medical history, drugs used for the treatment and drugs dosage, rehospitalization history, cardiac and non-cardiac disease records for the registered patients, in-hospital medical records as well as 6-month, 12-month, and 24-month follow-up records for the patients. First, we excluded the attributes containing the follow-ups information because our research focuses on the patients admitted in the hospital and stayed for medical examination regardless of the 6-month, 12-month, and 24-month follow-ups. We excluded 236 attributes containing follow-ups information from the total of 551 attributes. In the next step, we omitted the 185 attributes containing the unnecessary information such as date and time attributes, drugs attributes, drugs dosage attributes, treatment methods used during the hospital stays, and attributes containing the stents information such as stents size, diameter, length etc. Furthermore, we also removed all the features contains the information of our final predictions. As our final target variables are ACS outcomes such as ST-elevation myocardial infarction (STEMI) and non-ST-elevation myocardial infarction (NSTEMI), mortality prediction of the patients admitted in the hospitals due to heart problem such as CD and NCD, and prediction of discharge patients whether the patients are diagnosed and recovered from the ACS, died during the hospital stay, discharged hopelessly from the hospitals, or referred to the other hospitals due to some medical reasons. Therefore, we omitted all those variables which contains this medical information. Finally, we have the 13,104 patients’ record containing 125 important and useful attributes such as basic medical information (e.g., gender, age, height, weight, heart rate etc.), past medical history (e.g., chest pain, diabetes, hypertension, previous heart failure etc.), family medical history, information about medical diagnosis tools (e.g., electrocardiogram, Image Finding for MI, MRI, CT scan, ECHO etc.), medical findings (e.g., white blood cells, neutrophil, haemoglobin, platelets, glucose, creatinine, cholesterol etc.), PCI information, and initial diagnosis records etc.

We have mentioned the detailed explanation of the Data Extraction and Data Pre-processing in the section 2.2. Please have a look in the updated version of the manuscript.

Comment 6: Why didn’t you use the t-test to verify the significance and statistical analysis of the experimental dataset?

Response 6: Thank you so much for your comment. For analyses of our experimental dataset, we applied various statistical methods such as Chi-Square test and Analysis of Variance (ANOVA) test. In our dataset, we have different types of data such as categorical features and continuous features. To perform the statistical analysis and check their statistical significance, we applied statistical methods and calculated their significance. A chi-square test is used for categorical variables to show their relationship with target variables and interpret the discrepancies between the actual outcomes and expected outcomes. The ANOVA test is used to analyse the mean value differences and influence of independent variables on target variable. We used ANOVA test because it compares more than two groups to determine the relationship between them. In contrast, t-test is conducted when you have to find the population means between two groups.

The detailed explanation of Chi-Square and ANOVA is available in the updated manuscript. Please see the section “2.6. Statistical Analysis” for reference.

Comment 7: As mentioned in Table 2, Baseline Characteristics, some attributes have a high p-value and are statistically insignificant. Why did you include these attributes in your experiment?

Response 7: Thank you so much for pointing it out. In the experiment, we have subdivided the KAMIR-NIH dataset into 2 groups such as ST-elevation myocardial infarction (STEMI) and non-ST-elevation myocardial infarction (NSTEMI). We applied the Chi-Square test for categorical variables to calculate the percentages and frequencies in dataset, and ANOVA test for continuous variables to compute the means and standard deviations of the selected features. Most of the features have p-value < 0.001 and <0.05 which indicates that the attributes are statistically significant and have very low chances of being wrong, whereas some features have higher p-value indicating that there are higher chances of wrong values and statistically not significant. In the Table 2, the features like Sex, Age, Chest Pain etc. have p-value <0.001, whereas some features like hsCRP, ST Type etc. have p-value <0.05 which shows that these attributes are statistically significant and have very low chances of wrong results. In contrast, the features like Neutrophil, Troponin T, High Density Lipoprotein etc. have higher p-value indicating that these attributes are not statistically significant and have higher chances of being wrong. The reason to add the attributes with higher p-values and low significance is that these attributes are important in the medical domain and have real-time high effect on the patients affected with cardiovascular disease. These are the most important attributes which affects the mortality of cardiac patients during the hospital stay as well as after the discharge. Therefore, we added these important attributes in the experimental analysis and implementation of the machine learning and proposed ensemble predictive model.

Comment 8: The research limitations should be merged with the conclusion and removed the section 5.1.

Response 8: Thank you for your valuable suggestion and remarks. We have merged the research limitations into the conclusion section. Please see the Conclusion chapter in the updated manuscript. The updates are highlighted in yellow colour.

Reviewer 3 Report

The article is devoted to the development of a machine learning-based soft voting ensemble classifier (SVEC) for predictive modeling of acute coronary syndrome (ACS) outcomes such as STEMI and NSTEMI, reasons for discharge for patients admitted to hospitals, and types of death for affected patients. during your stay in the hospital. The study's relevance is justified by the fact that, at present, machine learning (ML) is a revolutionary and advanced technology widely used in the field of medicine and medical informatics, especially in the diagnosis and prognosis of cardiovascular diseases. The authors used the Korean Acute Myocardial Infarction Registry (KAMIR-NIH) data set, containing data on 13,104 patients with 551 features. After extracting and preprocessing the data, 125 useful features were used, and the SMOTETomek hybrid sampling method was applied to oversample the imbalance of minority class data. The SVEC proposed in this study applied three machine learning algorithms, such as random forest, complementary tree, and gradient boosting machine, to predictively model our target variables and compare them with the performance of all base classifiers. Experiments have shown that SVEC outperforms other machine learning predictive models in accuracy (99.0733%), accuracy (99.0742%), recall (99.0734%), F1 score (99.9719%), and area under ROC -curve (AUC) (99.9702%).

Despite the satisfactory quality of the article, some shortcomings need to be corrected.

  1. It is recommended to describe the data used for the experimental investigation in more detail.
  2. The authors split the data into 70 and 30 for training and testing. Such division should be grounded.
  3. The selection of tests and metrics for analyzing the data and results should be justified.
  4. Statistical machine learning methods applied by the authors should be briefly described in the Methods section.
  5. The novelty of the proposed method should be highlighted.
  6. The practical aspect of the proposed approach implementation should be discussed.

In summarizing my comments, I recommend that the manuscript is accepted after minor revision. 

Author Response

Manuscript ID: sensors-2171136

Manuscript Title: “A machine learning applied prediction model for identification of acute coronary syndrome (ACS) outcomes and mortality in patients during the hospital stay”

To: Sensors Editor

Re: Response to reviewers

Dear Ms. Charlotte Li and Respected Reviewers,

Thank you and your team for considering our manuscript, entitled: “A machine learning applied prediction model for identification of acute coronary syndrome (ACS) outcomes and mortality in patients during the hospital stay” and the tireless efforts spent in reviewing the paper.

We are thankful to you and reviewers for the perceptive comments and valuable suggestions to further improve our manuscript. We have complied with all the requirements in detail. We hope that, with these modifications and improvements based on the comments, the quality of our manuscript would meet the publication standard of Sensors.

We are uploading our (a) point-by-point response to the comments (below) (response to reviewers), (b) an updated manuscript with yellow highlighting indicating changes, and (c) a clean updated manuscript without highlights.

Thank you very much for your kind consideration.

Best regards,

Prof. Jong Yun LEE

Department of Computer Science,

Chungbuk National University (CBNU),

Cheongju Chungbuk, 28644 Korea.

Response to Reviewers’ Comments

Reviewer 3

The article is devoted to the development of a machine learning-based soft voting ensemble classifier (SVEC) for predictive modeling of acute coronary syndrome (ACS) outcomes such as STEMI and NSTEMI, reasons for discharge for patients admitted to hospitals, and types of death for affected patients. during your stay in the hospital. The study's relevance is justified by the fact that, at present, machine learning (ML) is a revolutionary and advanced technology widely used in the field of medicine and medical informatics, especially in the diagnosis and prognosis of cardiovascular diseases. The authors used the Korean Acute Myocardial Infarction Registry (KAMIR-NIH) data set, containing data on 13,104 patients with 551 features. After extracting and preprocessing the data, 125 useful features were used, and the SMOTETomek hybrid sampling method was applied to oversample the imbalance of minority class data. The SVEC proposed in this study applied three machine learning algorithms, such as random forest, complementary tree, and gradient boosting machine, to predictively model our target variables and compare them with the performance of all base classifiers. Experiments have shown that SVEC outperforms other machine learning predictive models in accuracy (99.0733%), accuracy (99.0742%), recall (99.0734%), F1 score (99.9719%), and area under ROC -curve (AUC) (99.9702%).

Despite the satisfactory quality of the article, some shortcomings need to be corrected.

Comment 1: It is recommended to describe the data used for the experimental investigation in more detail.

Response 1: Thank you so much for your recommendation to add more detailed description for the experimental dataset. In the section “2.1. Data Source”, we have provided the detailed description for the KAMIR dataset. For the experimental analysis and research studies, we use the Korea Acute Myocardial Infarction Registry (KAMIR) dataset which is a nationwide registry for Korean patients affected with heart related diseases. This is registered in 52 hospitals all over the Korea and has all patients’ record registered from November 2005 to December 2019. This KAMIR data registry has categorized the whole data into 4 groups based on the time of registered patients. KAMIR-I has all the patients’ data registered between November 2005 to December 2006, KAMIR-II has the patients’ record from January 2007 to January 2008, KAMIR-III also known as KorMI-I which has all medical information of the patients between February 2008 to March 2012, KAMIR-IV (KorMI-II) has the data from April 2012 to December 2015, and KAMIR-NIH has all the registered patients record from November 2011 to December 2019. The latest data of this registry is KAMIR-NIH, so we use the KAMIR-NIH data for the experimental analysis. KAMIR-NIH consists of 13,104 patients’ records with 2-year follow-ups after hospital discharge. This data contains 551 attributes of the registered patients such as patients past medical history, basic medical information, drugs prescribed and used, rehospitalization history, and cardiac and non-cardiac disease records for the registered patients. This dataset contains the patients’ information during the hospital stay as well as 6-month, 12-month, and 24-month follow-up records.

Please have a look on section “2.1. Data Source” and 2.2. “Data Extraction and Data Preprocessing” for the detailed description of the experimental dataset.

Comment 2: The authors split the data into 70 and 30 for training and testing. Such division should be grounded.

Response 2: We are very thankful for your valuable comment to improve the quality of the manuscript. In the experiments, we have divided the dataset into 70% of the training dataset and 30% test dataset. The training dataset (70%) is further subdivided into training and validation dataset using 5-fold cross validation. In the 5-fold cross validation, the training dataset is divided into 5 equal sub-groups of datasets and trained the model using 4 subgroups as training data and the 5th subgroup as test data during the training process. In the next step, the other subgroup is considered as test dataset and trained the model on rest of the 4 subgroups of dataset. This process is repeated for 5 iterations until all subgroups are used as test data during the training process of the model. After the training of the model, the 30% test dataset is used to validate the model efficiency.

The complete evaluation of the data division is described in the figure below.

We have added the detailed description of the data splitting into the updated version of the manuscript. Please have a look in section “2.4. Architecture of Proposed Predictive Modeling System”. The updates are highlighted in yellow colour.

Comment 3: The selection of tests and metrics for analyzing the data and results should be justified.

Response 3: Thank you for pointing out the tests and metrics for data analysis and results. First of all, after the feature selection from the KAMIR raw dataset, we applied multiple missing values imputation methods such as mean value imputation, median, k-nearest neighbours, and also used the zero-value imputation to deal with missing values in the selected attributes. We compared and confirmed that the zero value imputation methods work efficiently than mean or median value imputation, so we finalized the zero-value imputation method to deal with missing values. For categorical features, we used the one-hot encoding [37] and label encoding [38] to convert them into numerical form. For continuous variable and discrete variables, we used the actual values of these attributes so that we can minimize the data loss. Furthermore, the variables containing multiple values, we used the one-hot encoding method so that we can use all the possible outcomes of those attributes. For binary-valued attributes, we simply converted them into 1 and 2 such that 1 represents Yes and 2 represents No, whereas if there are missing values, denoting them with 0.

After that we applied the SMOTETomek hybrid sampling technique which is valid data sampling technique according to the previous studies [39-41] and also from our experiments. After that we applied 70~30% data splitting method and 5-fold cross validation on our dataset. As as our dataset consists of categorical and continuous variables, we applied the chi-square test for categorical variables to calculate the percentages and frequencies in dataset, and ANOVA test for continuous variables to compute the means and standard deviations of the selected features.

             For the experiment, we applied multiple machine learning algorithms and choose the top 3 algorithms with outperformed results. After that, we proposed the soft voting ensemble classifier as predictive model for the prescribed outcomes and compared the results of the base classifiers with the proposed SVEC using performance measures such as precision, recall, F1-score, AUC, and accuracy. The experimental results have verified the performance and authenticity of the proposed soft voting ensemble classifier for the predictive modelling of the target variables. The results of the proposed ensemble model are outperformed except the AUC value, which is higher for Extra Tree classifier.

For the performance evaluation of the proposed soft voting ensemble classifier, we can use ROC curve or confusion matrix. ROC curve can be used when we have binary variables or the output consists of only 2 classes such as “Yes and No” or “1 and 0” etc. As our target variables are 10 such as “STEMI CD”, “STEMI NCD”, “STEMI Hopeless Discharge”, “STEMI Recovery to Home”, “STEMI Recovery to Other Hospital”, “NSTEMI CD”, “NSTEMI NCD”, “NSTEMI Hopeless Discharge”, “NSTEMI Recovery to Home”, and “NSTEMI Recovery to Other Hospital”, we drew the confusion matrix for the performance evaluation. It is represented in tabular form in which x-axis and y-axis represents the class labels which are denoted as numbers from 0-9 for these ten class labels, respectively.

We have described all the above-mentioned details in the manuscript. Please have a look on the highlighted version of the manuscript.

Comment 4: Statistical machine learning methods applied by the authors should be briefly described in the Methods section.

Response 4: Thank you for your valuable remarks. In the revised version of the manuscript, we have added more contents for the applied machine learning models as per your suggestions. Please see the section “2.5. Applied Machine Learning Algorithms” in the updated version of the manuscript. The updates are highlighted in yellow colour.

Comment 5: The novelty of the proposed method should be highlighted.

Response 5: Thank you so much for highlighting the important point to update the manuscript.

As per your suggestion, we have highlighted the novelty of this research work in the introduction of the manuscript and also included the research contributions and practical aspect of the proposed approach. Please see the Introduction section in the updated version of the manuscript.

Comment 6: The practical aspect of the proposed approach implementation should be discussed.

In summarizing my comments, I recommend that the manuscript is accepted after minor revision.

Response 6: Thank you so much to mention and let us include the practical aspect of this research in the manuscript. It is very important to add the real-time use of the proposed approach in health care. We have discussed the practical use of the proposed method in the Introduction and Discussion of the manuscript. Please have a look on the highlighted part of the Introduction and Discussion.

We really appreciate your efforts to review the manuscript and highlighted the important points to improve the quality of the research article. We are pleased to get the acceptance from you with minor revisions. I have updated the manuscript accordingly and provided the detailed explanation of your comments in the manuscript and response letter.

Round 2

Reviewer 1 Report

The authors have addressed my comments adequate and accordingly. I have no further comments and would recommend this manuscript for publication.

Reviewer 2 Report

The author incorporated changes based on my suggestions and review, resulting in a polished and improved final draft.